# Foraging through multiple nest holes: An impediment to collective decision-making in ants

**Marine Lehue\*, Claire Detrain**

Unit of Social Ecology (CP.231), Université Libre de Bruxelles, Brussels, Belgium

\* marine.lehue@ulb.ac.be

## Abstract

In social insects, collective choices between food sources are based on self-organized mechanisms where information about resources are locally processed by the foragers. Such a collective decision emerges from the competition between pheromone trails leading to different resources but also between the recruiting stimuli emitted by successful foragers at nest entrances. In this study, we investigated how an additional nest entrance influences the ability of *Myrmica rubra* ant colonies to exploit two food sources of different quality (1M and 0.1M sucrose solution) and to select the most rewarding one. We found that the mobilisation of workers doubled in two-entrance nests compared to one-entrance nests but that ants were less likely to reach a food source once they exited the nest. Moreover, the collective selection of the most rewarding food source was less marked in two-entrance nests, with foragers distributing themselves evenly between the two feeders. Ultimately, multiple nest entrances reduced the foraging efficiency of ant colonies that consumed significantly less sugar out of the two available resources. Our results highlight that the nest structure, more specifically the number of nest entrances, can impede the ant's ability to process information about environmental opportunities and to select the most rewarding resource. This study opens new insights on how the physical interface between the nest interior and the outside environment can act upon collective decision-making and foraging efficiency in self-organized insect societies.

## 1 Introduction

Group-living animals have to share common goals, such as finding a suitable nesting place or exploiting a profitable food resource. However, the pay-offs of a collective decision obviously depend on their ability to integrate multiple–and sometimes conflicting–information sources in order to select the best option for the group as a whole [1–3]. In such situations where group coordination is beneficial, theoretical studies have demonstrated that pooling different sources of information in order to converge toward a shared decision could be more advantageous to all group members (as they are more likely to be correct) than decisions made by a few leaders [4–8].

**Data Availability Statement:** All data files are available in the Zenodo Digital Repository: https://zenodo.org/record/3803696#.XtUtEcA693g. The DOI for the data is: 10.5281/zenodo.3803696.

**Funding:** M.L. was supported by a Belgian PhD Grant from the F.R.I.A. (Fonds pour la formation à la Recherche dans l'Industrie et dans l'Agriculture). C.D. is Research Director from the Belgian National Fund for Scientific Research (FRS-F.N.R.S). This paper was published with the Financial support of the "Fondation Universitaire de Belgique " and of the grant N°CDR J.0053.18F from FRS-FNRS. The funders had no role in study design, data collection and analysis, decision to publish, or preparation of the manuscript.

**Competing interests:** The authors have declared that no competing interests exist.

Group-level coordination may occur through self-organising processes, during which complex collective behaviours emerge through multiple and simple interactions at the individual level. In such cases, all group members follow their own behavioural rules, rely on local information, local communication and local reaction to neighbouring individuals. Individual responses are regulated through positive and negative feedback processes that amplify or dampen the emergent group behaviours [9–11]. The overall result is a coordinated behaviour and that, in most cases, allows for the best choice among several options. Self-organized processes have been evidenced across several taxa including humans [10–15]. Insect societies offer among the most compelling examples of self-organized adaptive choices, such as the selection of the best nesting site [16,17], the use of the shortest path between the nest and a resource [18], or the selection of the best food source [9, 19–21]. These complex collective behaviours can emerge without requiring high cognitive abilities or global overview of the group by the colony members.

In insect societies, collective foraging relies on the active recruitment of nestmates inside the nest. In honeybees, recruiters perform a waggle dance in the nest to mobilize recruits and to indicate the spatial location of a patch of flowers [22]. In many ant species, food recruitment is initially triggered by the scouts that have successfully discovered a food source and that lay a recruitment trail when returning to the nest. Pheromone trails coupled to antennal contacts displayed by recruiting ants in the nest will stimulate and guide new ants to the food source [9, 23]. In this process, a tuning of signaling has evolved to bias the colony choice towards the most valuable option. For instance, bee or ant foragers can tune the intensity of their recruitment signal according to food quality. Bee recruiters will perform longer-lasting and more intense dances [21,24], while ants will deposit larger amount of trail pheromone towards high-quality resources [19, 25, 26–28]. Through their higher investment in recruiting signals, the individuals that discovered richer sources will thus drive the group's choice toward the best option, even though each recruiter does not directly determine the resource that will be ultimately selected by the colony.

Collective decision-making may therefore benefit from the convergence of "informed" individuals at a single place where nestmates can compare multiple signals differing in their quality and/or intensity. In ants, the selection of the best resource is facilitated when the pheromone trails, of which the concentration is correlated to the resource quality, converge toward a single point where the different options can be easily compared by nestmates. In natural conditions, the key location at which information can be compared is the main entrance of the ant nest, where interactions between returning foragers and inner workers occur [29–34].

Additional nest entrances will increase the number of potential sites where recruitment and information sharing take place. Because information no longer converges to a single location, the synchronization of foraging activity may be more difficult to achieve, and signals may become locally weaker, thus preventing the emergence of a collective response. In a previous study on *Myrmica rubra* ants, we found that an additional nest entrance segregated the pool of recruiters and hampered the formation of a collective foraging trail leading to a food source [31]. Here, we hypothesise that spatial constraints on how the information can be shared among group members, will greatly influence the pay-offs and the accuracy of collective decision-making. More precisely, we investigate whether and how the addition of a second nest entrance to *Myrmica rubra* colonies may influence their ability to collectively exploit and discriminate between two food sources of different sucrose concentration (1M and 0.1M). We will compare the foraging efficiency in terms of workers' mobilisation, collective choice of the high quality food source and sucrose consumption for the same ant colonies when being kept in either a one- or a two-entrance nest.

## 2 Material & methods

### 2.1. Ant colonies

*M. rubra* is a polygynous and monomorphic ant species that is common in European temperate areas. Its natural nests show from a single up to six active entrances, with some being aggregated into clusters with a between-entry distance of 5cm on average (personal observations). *M. rubra* nests are typically composed from several hundred to 1,500 workers (based on our personal observation and [35]). For the nests that were dug under stones or under wood logs, the superficial nest chambers housed a few hundred individuals and consisted of a large single chamber or of multiple chambers, separated by loose walls or well- defined ridges (personal observations). Nine *M. rubra* colonies were excavated from earth banks in a semi-open grassland located in Aiseaux and Falisolle (E 004˚35.703', E 004˚37.915'; Belgium) in June 2016. Once in the laboratory, ant colonies were reared in test-tube nests covered with a red filter and placed in foraging arenas with Fluon-coated walls (Whitford, UK) to prevent ants from escaping. We kept laboratory conditions at 21 +/- 0.4 C˚ and 52 +/- 2% relative humidity, with a constant photoperiod of 12 hours a day. Ants were fed with water and sucrose solution (0.3M) ad libitum and with mealworms twice a week.

### 2.2. Experimental setup

Experimental nests were made out of a laser-cut Plexiglas circular wall covered with a Plexiglas ceiling. Internal dimensions of the circular nests were 8-cm diameter and 2-mm-high. Each of the nine experimental colonies contained one queen, 300 workers and brood covering around 10% of the nest area. The nest comprised three entrances (each 10mm wide and 5mm long) that were placed 15 mm apart from each other and that could be close or open depending on the tested nest configuration (Fig 1). We used two different nest configurations with either one open entrance (i.e. the central entrance) or two open entrances (i.e. the two lateral entrances). We used fitted pieces of cardboard to close the nest entrances. For the two-entrance configuration, entries were thus separated by 3 cm which is a value close to the one observed in natural nests (personal observations). We placed the experimental nest on one side of a rectangular arena (45 x 30 cm) as shown in Fig 1. We covered the floor of the arena with plaster and daily watered around the nest to provide the humidity necessary to the ant survival. Before the start of an experimental series, we moved ant colonies into these experimental nests, where they could acclimatize for 48h.

### 2.3. Experimental procedure

We tested whether the number of nest entrances can influence the ants' collective choices between food sources that differed in their quality (here in their sugar concentration). To do so, we tested each colony in a one-entrance-nest and in a two-entrance-nest in a pseudo-randomized order. An experimental series was carried out as follows. First, in order to stimulate recruitment, we deprived the ants of sugar and protein for 48h. The feeders consisted of circular plates (3-cm diameter) with a central sugar-filled reservoir of which the cross-shape increased the perimeter/area ratio, thereby reducing congestion effects around the food droplet. On the experimental day, we placed in the arena the two feeders, each offering 600 μL of either 1M or 0.1M sucrose solution. We placed the two feeders each at 25 cm from the central entrance and 8 cm apart from each other (Fig 1). The experiment lasted 120 minutes during which we filmed the entire arena using Logitech C920 webcams (1920x1080 pixel resolution, 15 fps). At the end of the experiment, we removed the feeders and changed the nest configuration by opening/closing the entrances. Colonies rested for four days in the new nest

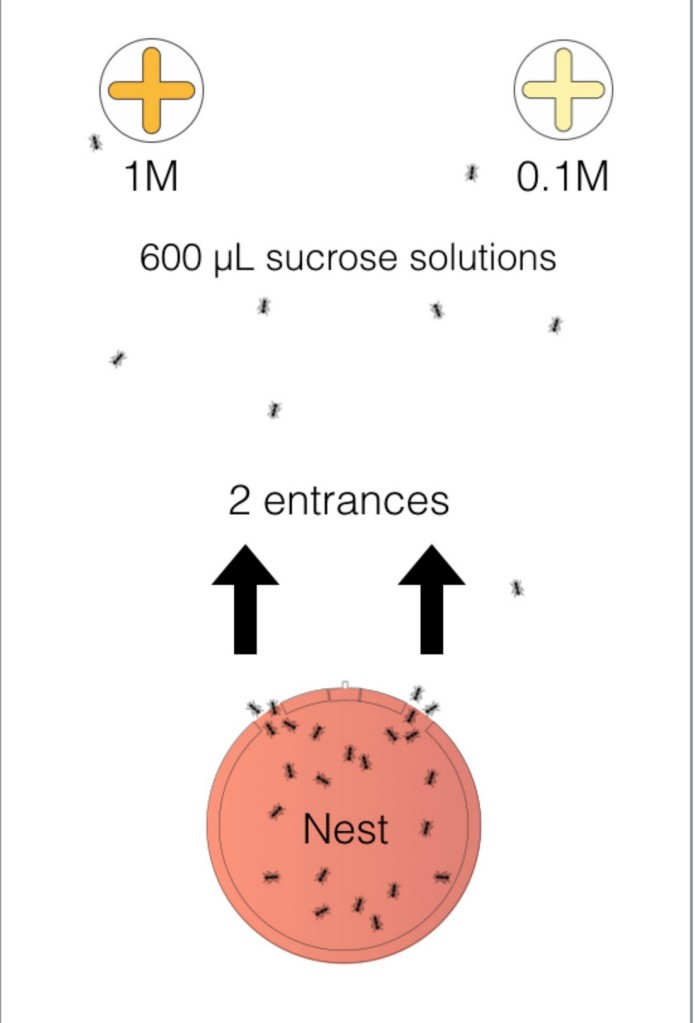

**Fig 1. Experimental set-up.** Colonies were housed either in one-entrance nests or two entrance nests. Nests were placed on one side of the arena and two feeders containing 600uL of either 1M and 0.1 M sucrose solution were equidistantly put on the opposite side.

configuration before undergoing the second experiment. This 4-days period provided enough time for ants to dynamically reorganize themselves inside the nest and for the flows of foragers to be equally spread between all the open entrances [30]. During this resting period, each colony could freely explore the foraging area and had access *ad libitum* to water, a 0.3M sucrose solution, and *Tenebrio molitor* mealworms.

## 2.4. Mobilisation of workers

We assessed the level of mobilisation of foragers in each nest configuration. First, at the beginning of each experiment, we measured the density of ants located in the entrance area, as these nestmates were the most likely to interact with incoming foragers [30]. In this type of artificial nests, the entrance area (5.6 cm$^2$) corresponded to a two-centimetre radius centred on the nest opening [30]. Once we introduced the food source, we counted, every 5 minutes, the number

of ants staying on each of the 3-cm diameter feeder plate. Concurrently, we measured the out-flow of ants per 5 minutes in order to obtain the total number of mobilized workers for the whole duration of the experiment. For technical reasons, we video-recorded the outflows in only seven colonies out of nine for both nest entrance configurations. The ant densities as well as the total number of mobilized ants were compared between the two nest-configurations using Wilcoxon signed-rank tests. We used a two-way ANOVA for repeated measures to test the effects of nest configuration (one-entrance or two-entrance) and time interval on the flows of outgoing workers.

For the two-entrance nest configuration, we characterised the distribution of the total out-flow of ants between the two entrances. To this aim, we computed an index of asymmetry $I_a$ as follows:

$$I_a = \left| \frac{F_L - F_R}{F_L + F_R} \right|$$

with $F_L$ and $F_R$ being the total outflow of ants through the left and right entrance respectively. This index varies between 0 for a perfectly symmetrical use of both entrances and 1 for a totally asymmetrical use of a one entrance by outgoing ants.

## 2.5. Efficiency at reaching the food source

To investigate whether and how a supplementary entrance influences the efficiency of ants at reaching and exploiting food sources, we performed an individual tracking of recruited individuals on the foraging area. For each colony, the tracking of foragers started 30 minutes after the food introduction, once the recruitment was well-established. Twenty ants exiting each open entrance were individually followed for a maximum duration of three minutes. As for the outflows, we examined only seven colonies out of nine for both nest entrance configurations, which resulted in the tracking of 140 ants in one-entrance nests and 280 ants in two-entrance nests. To avoid a possible bias in trail-following due to knock-on effects among ants that simultaneously exited the nest, we tracked 1 ant every 5 outgoing ants. At the end of the three-minute observation, the ant could have reached the 1M food source, reached the 0.1M food source, gone back to the nest, or kept on strolling in the nest surroundings. We compared the proportion of ants in each of these categories for the two nest configurations by using a chi-square test. For the population of ants that reached feeders, we tested whether they were equally distributed between the two feeders by using a binomial test with a probability of 0.5. For each experiment, 30 minutes after food introduction, five ants that had reached a feeder were randomly chosen and we measured whether they decided to drink the food solution as well as the duration of their drinking behaviour. At least three minutes elapsed between successive observations of ant individuals at the feeders. The percentage of drinking ants as well as the duration of their feeding behaviour were compared between the two nest configurations by using a Chi-square test and a Mann-Whitney test, respectively.

## 2.6. Sucrose consumption and relative exploitation of the two food sources

The global efficiency of food exploitation was assessed by measuring the ants' consumption at the two sucrose solutions. Food plates were weighted using a microbalance ($10^{-5}$ g accuracy, Metler Toledo AB125-S) three times: empty, just after adding the 600uL of sucrose solution at the start of the experiment, and after food consumption by the ants at the end of the experiment. We considered the evaporation rate of the sucrose solutions by placing two control food sources of each concentration (same volume, 1M and 0.1 M) next to the experimental arenas and by weighing them at the end of the experiment. The evaporation rates were calculated and

taken into account to quantify the sucrose solution that was actually ingested by the ants. To limit possible spatial bias on the level of food exploitation, we placed the most concentrated food source alternatively either on the left or the right side of the arena. As the experiments were paired per colony, the total sucrose consumption and sucrose consumption at each feeder were compared between the two nest configurations using Wilcoxon signed rank tests (two-tailed tests).

In addition, the dynamics of food exploitation was obtained by counting the number of ants present at each food source, every five minutes for the whole duration of the experiments (120 min). We used two-way ANOVA's for repeated measures to test for the effects of nest configuration and time interval on the occupancy of feeders by ant foragers. We also computed an index of asymmetry of food exploitation based on the distribution of the foragers between the two available food sources. The index of asymmetry $I_a$ was calculated as follows:

$$I_a = \frac{n_{1M} - n_{0.1M}}{n_{1M} + n_{0.1M}}$$

with $n_{1M}$ and $n_{0.1M}$ being the number of foragers at the 1M and 0.1M feeder respectively. This index varies between -1 (all foragers located at the 0.1M food source) to 1 (all foragers located at the 1M food source).

## 3. Ethic statement

No licences or permits were required for this research. Ant colonies were collected with care in the field and were maintained in nearly natural conditions in the laboratory. Ants were provided with suitable nesting sites, food and water, thus minimizing any adverse impact on their welfare. After the experiments, the rest of the colony was kept in the laboratory and reared until their natural death.

## 4. Results

### 4.1. Mobilisation of workers

Prior to the experiments, the densities of ants at the entrances, which could have influenced the further recruitment of nestmates, were not significantly different between one or two-entrance nests (respectively 3.08±1.54 ant.cm$^{-2}$, $n = 9$, and 2.14±1.28 ant.cm$^{-2}$, $n = 7$ Wilcoxon signed rank test, W = 18, $p = 0.15$).

We found that the outflow of foragers exiting the nest increased with the number of nest entrances. Indeed, after two hours of food exploitation, the total number of mobilized foragers in two-entrance nests was twice as high as in one-entrance nests (mean±SD, 836 ants±259 vs 467±121 respectively, $n = 7$, Wilcoxon signed rank test, $p = 0.031$, Table 1). The outflows of

**Table 1. Foraging efficiency of ant colonies kept in one-entrance and two-entrance nests.**

|  |  | One-entrance nests | Two-entrance nests | P value | Wilcoxon signed-rank test |
|---|---|---|---|---|---|
| Mobilization mean±SD | Total outflow (N Ants) | 467±121 ($n = 7$) | 836±259 ($n = 7$) | 0.031* | $W = 26$ |
|  | Total solution ingested (mg) | 112±20 ($n = 9$) | 99±26 ($n = 9$) | 0.12 | $W = 27$ |
| Ingested solution mean±SD | 1M solution ingested (mg) | 85±15 ($n = 9$) | 59±21 ($n = 9$) | 0.019* | $W = 39$ |
|  | 0.1M solution ingested (mg) | 27±15 ($n = 9$) | 40±10 ($n = 9$) | 0.024* | $W = 39$ |
|  | Total weight ingested (mg) | 30.0±5.2 ($n = 9$) | 21.5±7.1 ($n = 9$) | 0.027* | $W = 37$ |
| Ingested Sucrose mean±SD | Weight ingested from 1M feeder (mg) | 29.1±5.1 ($n = 9$) | 20.2±7.2 ($n = 9$) | 0.019* | $W = 39$ |
|  | Weight ingested from 0.1M feeder (mg) | 0.9±0.5 ($n = 9$) | 1.4±0.3 ($n = 9$) | 0.024* | $W = 39$ |
| Sugar Yield mean±SD | Sugar weight ingested per mobilized ants (mg.ant$^{-1}$) | 0.064±0.012 ($n = 7$) | 0.024±0.007 ($n = 7$) | 0.016* | $W = 28$ |

ants steeply increased during the first steps of food recruitment and then progressively decreased over the course of the experiment. The 5-minute outflows were also influenced by the number of nest entrances (Fig 2: Two-way ANOVA with repeated measures: nest configuration effect: $F_{1,288} = 11.64$; $p<0.01$, time effect: $F_{23,288} = 9.76$, $p<0.001$, interaction effect: $F_{23,288} = 0.61$, $p = 0.92$). Even though the same total amount of food was made available, the number of entrances had thus a deep impact on the recruitment of nestmates, leading to the doubling of the mobilisation of workers in two-entrances nests. We never observed any structured foraging trail emerging from holes, regardless of nest configuration.

In the two-entrances nests, we also compared the mobilization of workers through each of the two open doors. The index of asymmetry $I_a$ ranged from an almost perfectly symmetrical use of the two entrances, with a minimal value of $I_a = 0.009$, to an asymmetrical use of a preferred entrance, with a maximal value of $I_a = 0.794$. When the colonies used nest entrances in a highly asymmetrical way, the choice of the favoured entrance was not related to its proximity to the richest food source. Indeed, the most used entrance was the one located on the same side as the 1M food source for only three out of the five colonies that showed a high level of asymmetry $I_a> 0.100$.

## 4.2. Efficiency at reaching the food source

Although the ants' mobilisation doubled in two-entrance nests, the number of foragers that reached a food source was strikingly similar for the two nest configurations. Indeed, we found that the total number of ants present at the two food sources changed over time but was not influenced by the number of nest entrances. (Fig 3: Two-way ANOVA with repeated measures: nest configuration effect: $F_{1,400} = 0.03$, $p = 0.87$, time effect: $F_{24,400} = 17.3$, $p<0.001$, interaction effect: $F_{24,400} = 0.69$, $p = 0.86$). In the early stages of the experiment, a slightly higher number of ants were present at the food sources for two-entrance nests but this difference quickly vanished over the course of the experiment (Fig 3). As we did not observe any cluster of ants that

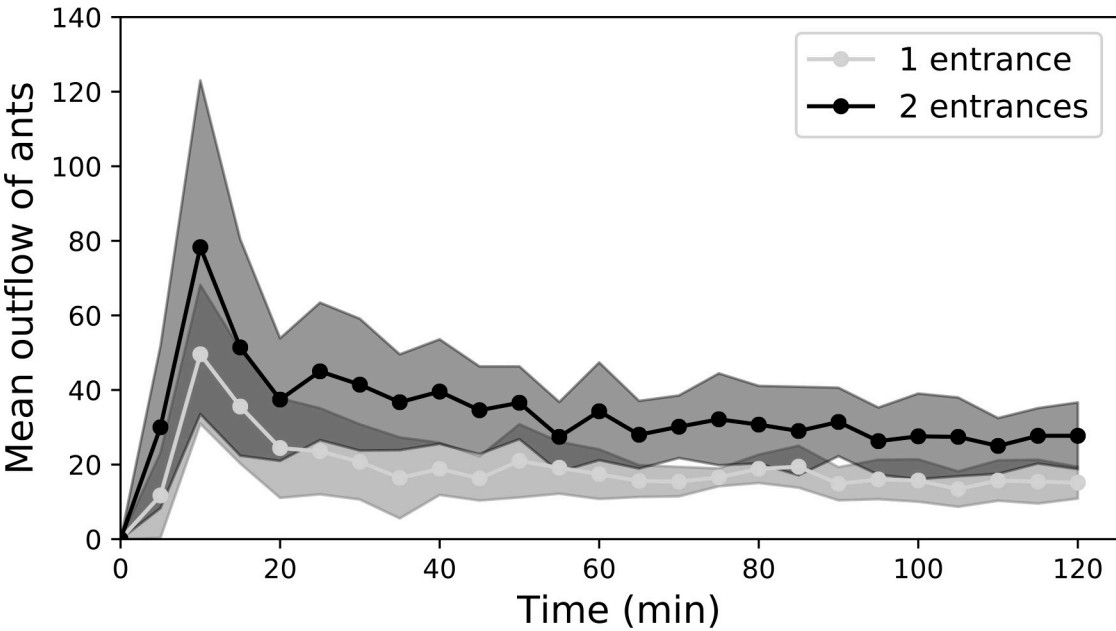

**Fig 2. Dynamics of ants' mobilisation out of a one-entrance or a two-entrance nest.** Flows of ants outgoing from one-entrance nests and two-entrances nests are represented every 5 min in grey and black respectively. Circles and shadings represent the mean ± SD, respectively ($n = 7$).

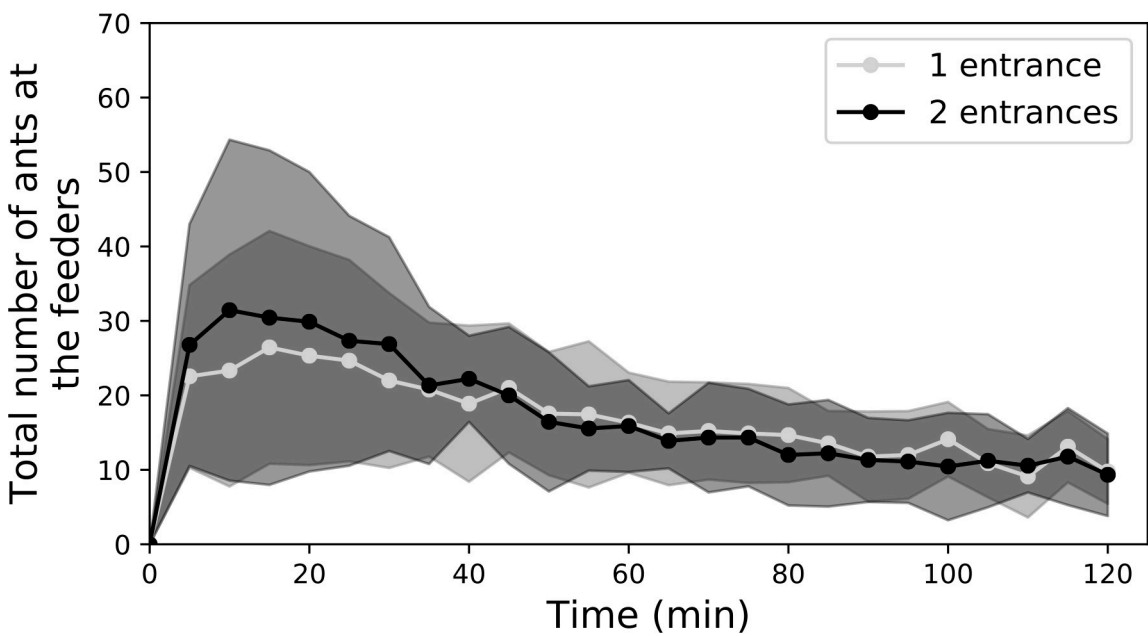

**Fig 3. Dynamics of the total number of ants at the two feeders.** The number of ants at each feeder was measured every five minutes over the course of the experiment, for one-entrance nests and two-entrance nests (in grey and black respectively). Circles and shadings represent the mean±SD, respectively ($n$ = 9).

might have hampered the reaching of the food source by nearby workers, this suggests that ants were less efficient at reaching the food sources during the first steps of recruitment from a two-entrance nest.

Thus, we individually tracked foragers once the recruitment was established for all colonies (i.e. after 30 minutes of experiment). Once ants had exited the nest, their probability to reach a food source was influenced by the number of nest openings. In the case of one-entrance-nests, we found that, within a 3-minutes period of observation, 43% of ant individuals reached a food source, 21% went back to the nest, and that 36% kept on strolling in the arena ($n$ = 140, Fig 4). In the case of two-entrance nests, a smaller proportion of ants (34%) reached any of the two food sources (Chi-square test, $n_1$ = 140, $n_2$ = 280, $p$ = 0.003, df = 3, $\chi^2$ = 13.7, Fig 4). Out of this nest configuration, the majority of mobilised ants went back to the nest (38%) and fewer ants remained exploring the environment (28%, $n$ = 280). Such a higher proportion of ants going back to the nest indicates a reduced ability of recruited ants to follow the pheromone trails laid by nestmates towards the feeders. For the ants that succeeded in reaching a food source, the slight differences in the Euclidian distances to the food sources between one- or two-entrance nests had negligible impact on the duration of the foraging journeys. Indeed, in one-entrance and two-entrance nest conditions, the average trip duration towards the feeder were respectively of 82 (SD: ±34) seconds and 89 (SD±40) seconds to reach the 1M food source ($n_1$ = 40, $n_2$ = 55, Mann-Whitney U test, $p$ = 0.42). Likewise, the trip duration to reach the 0.1M food source were respectively of 97 (SD±52) and 96 (SD±42) seconds at one-entrance and two-entrance nests ($n_1$ = 19, $n_2$ = 41, Mann-Whitney U test, $p$ = 0.70).

Furthermore, we examined the influence of nest configuration on the ability of ants to reach the most rewarding food source. Based on data of individual tracking, we found that among all the ants that exited the one-entrance nest and that reached a food source (60 out of 140 ants), a significantly larger proportion of ants (68%) reached the 1M food source than the

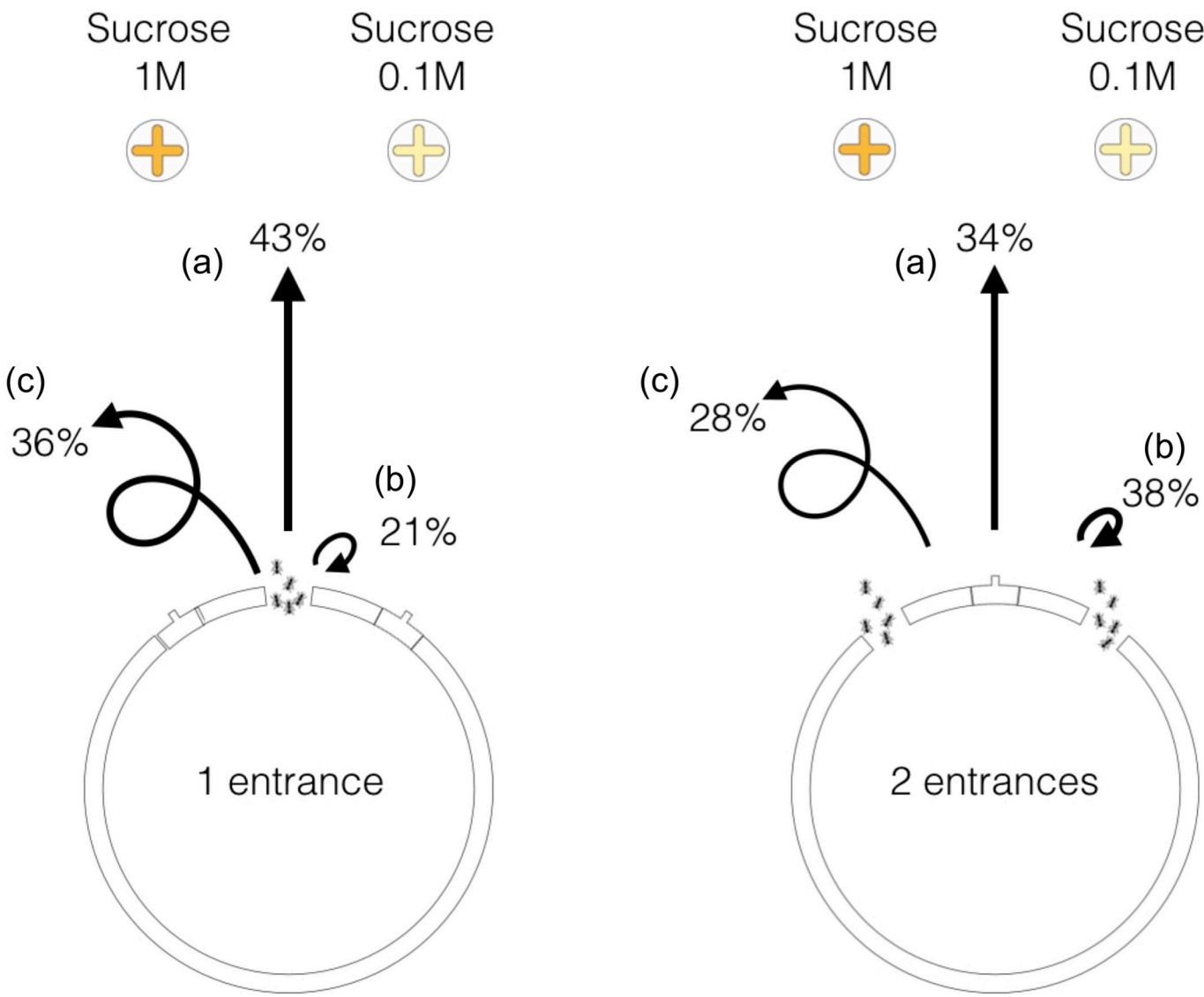

**Fig 4. Influence of the nest entrance configuration on the ant's journey outside the nest.** Proportion of ants reaching any food source (a), going back to the nest (b) or remaining in the arena (c) after 3 minutes of observation (n = 140 for one-entrance nests, and n = 280 for two-entrance nests).

0.1M feeder (41 out of 60 ants, all colonies pooled, binomial test, $p$ = 0.006, Fig 5). By contrast, in two-entrance nests, the foragers that reached a food source (96 out of 280 ants) were as likely to reach the 1M feeder than the 0.1M (respectively 58% and 42% out of 96 ants, all colonies pooled, binomial test, $p$ = 0.12, Fig 5). At the level of each entrance, ants exiting from the entrance located on the same side as the 0.1M feeder had the same probability to reach the 1M feeder as the 0.1M one (51%, 22 out of 43 ants, binomial test, $p$ = 1, Fig 5). For the ants exiting the entrance located on the side of the 1M feeder, the proportion of workers that reached this feeder (64%, 34 out of 53 ants, Fig 5) was slightly higher, although not significantly different from a random distribution (binomial test with an equal probability of 0.5 to reach each feeder, $p$ = 0.053). This result suggests that ants were less able to efficiently compare competing trails leading to sources of different quality when they exited from a nest with multiple openings.

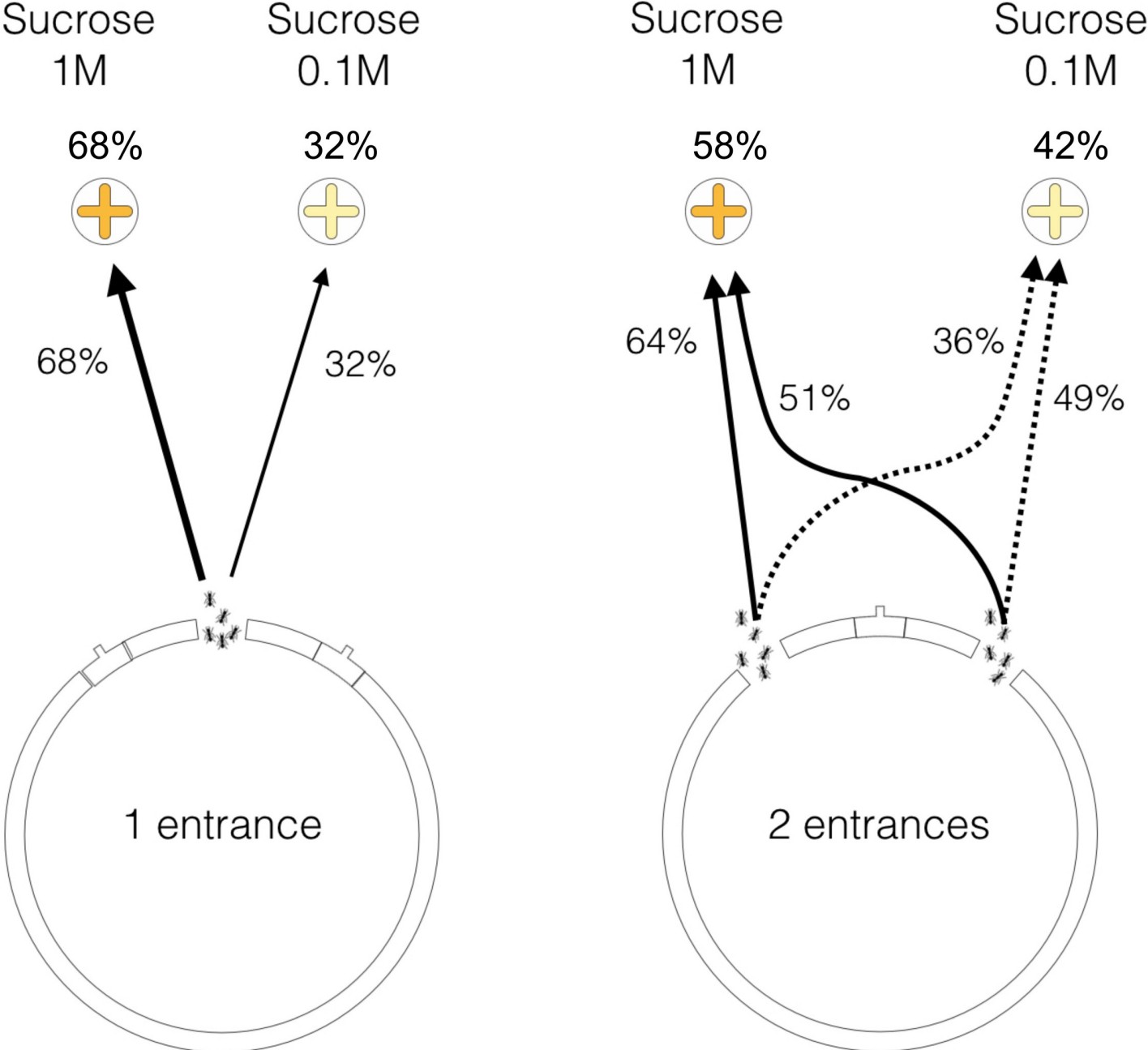

**Fig 5. Influence of the nest entrance configuration on the selection of food sources.** Among the ant population that reached a food source, the figure shows how ants distribute themselves between each of the two food sources (1M and 0.1M). For one-entrance nests $n = 60$, for two-entrance nests $n = 96$.

## 4.3. Relative exploitation of the two food sources and sucrose consumption

The population of foragers at each feeder increased over the course of the experiment and was influenced by the food quality when ants were recruited from a one-entrance nest. (Fig 6A: Two-way ANOVA for repeated measures, food quality effect: $F_{1,400} = 19.02$, $p<0.001$, time effect: $F_{24,400} = 9.44$, $p<0.001$, interaction effect: $F_{24,400} = 3.0$, $p<0.0001$). From the start of the experiment, the 1M feeder was more exploited than the poorer 0.1M food source. This preference was amplified over time leading to a majority of workers exploiting the 1M feeder for the

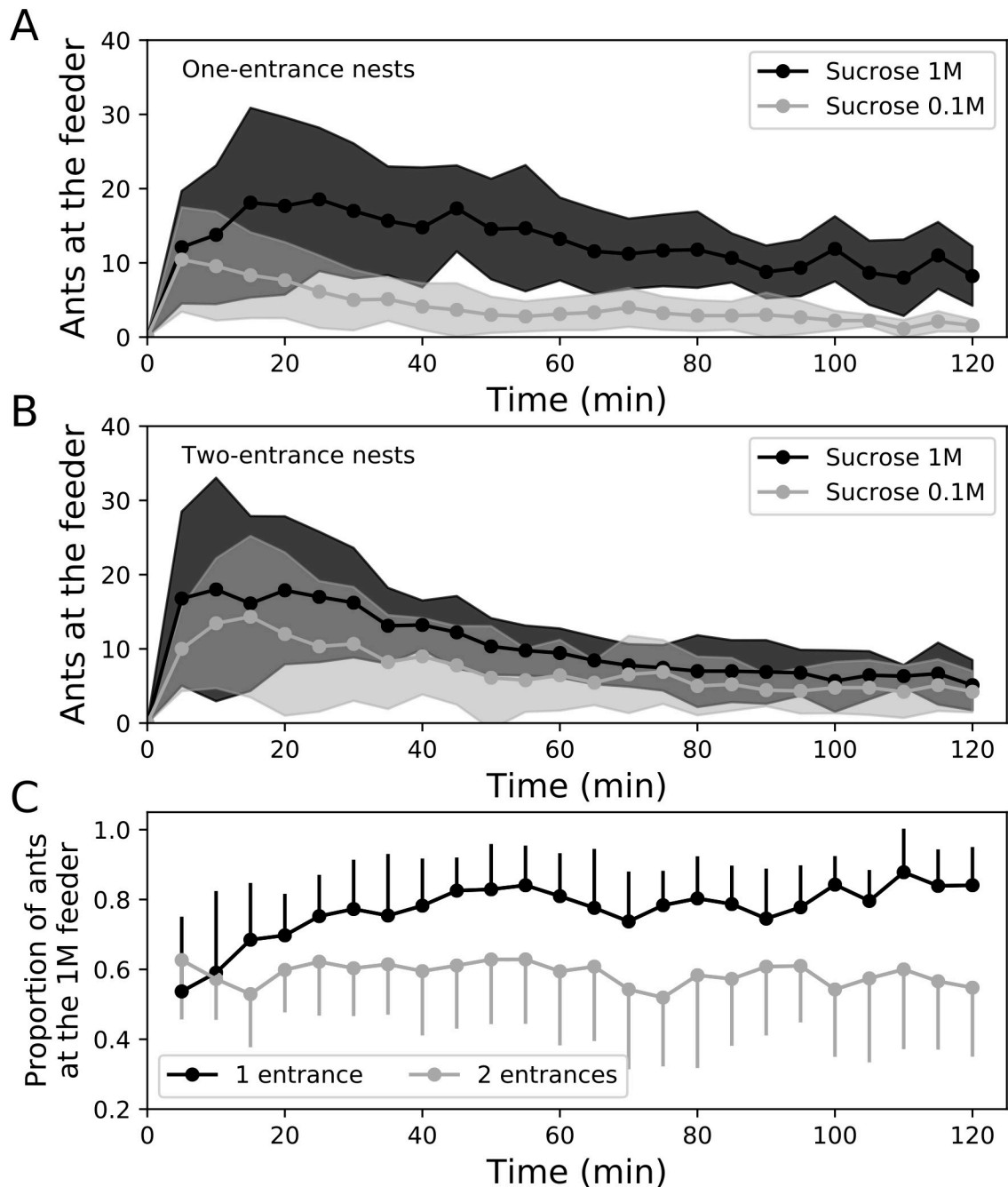

**Fig 6. Relative exploitation of feeders over time.** Number of ants at the 0.1M or the 1M feeder for (A) one-entrance nests, and (B) two-entrance nests as a function of time. Proportion of ant present at the richest feeder in both nest entrance configurations (C). In each experiment, one feeder was filled with 1M sucrose solutions (black circles, dark grey shading) and the other feeder with 0.1M sucrose solution (light grey circle, light grey shading). Circles and shadings represent the mean ± standard deviation, respectively.

one-entrance nest condition (Fig 6C). For the two-entrance nest condition, the population of foragers at the food source changed over time but in a similar way at each feeder, regardless of its sugar concentration (Fig 6B: Two-way ANOVA for repeated measures, time effect: $F_{24,400} = 13.46$, $p<0.001$, food quality effect: $F_{1,400} = 2.11$, $p = 0.17$, interaction effect: $F_{24,400} = 0.79$, $p = 0.75$). In accordance with the former results of individual tracking, the proportion of

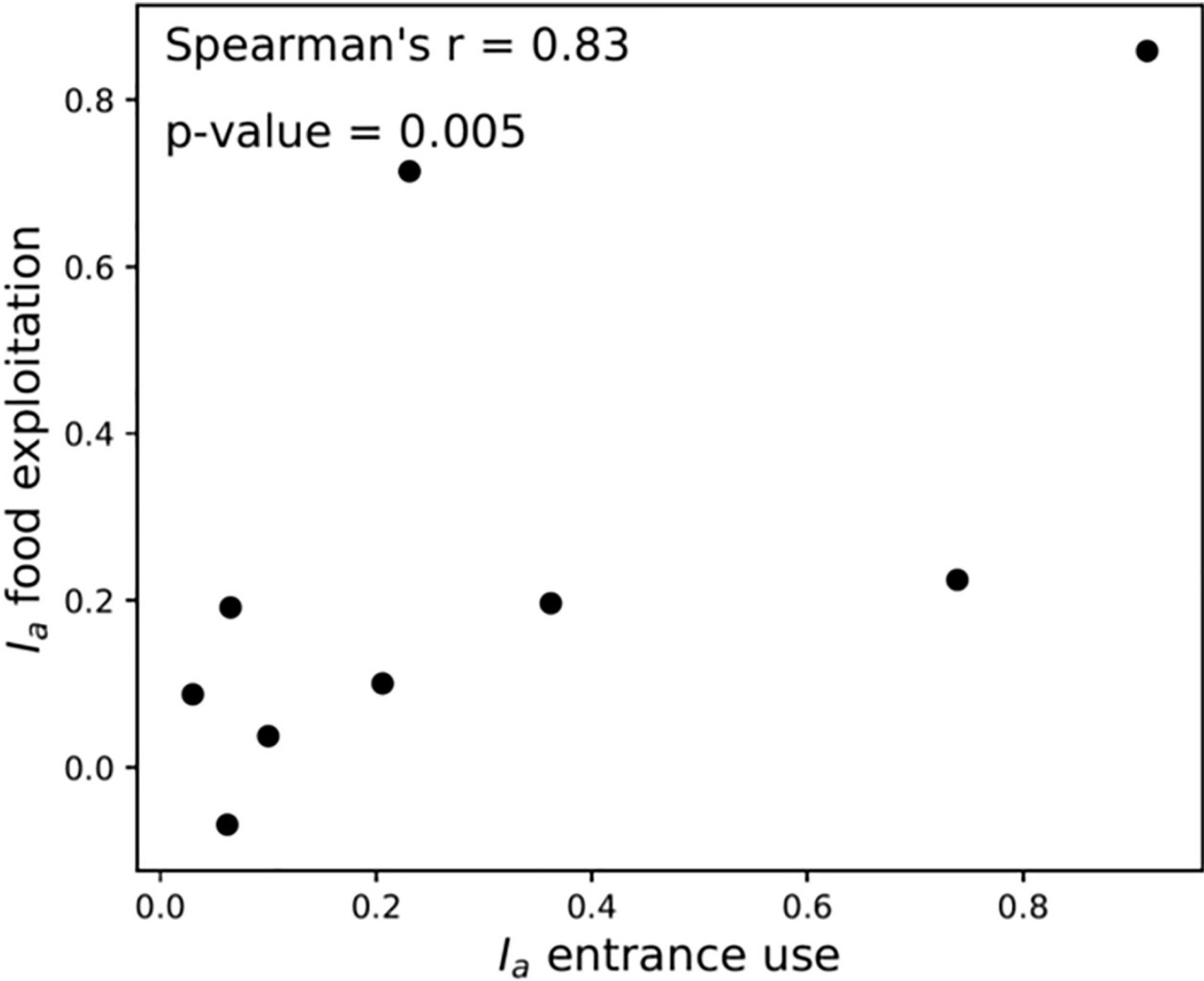

**Fig 7. Asymmetry in the exploitation of feeders is correlated to asymmetry in the entrance use.** Asymmetry in the entrance use ranged from 0 (symmetrical use of entrances) to 1 (use of only one entrance) after 20 min of experiment. Asymmetry of the resource exploitation was measured at the end of the experiment and ranged from 1 (all ants at the 1M source) to -1 (all ants at the 0.1M source). Spearman's correlation, n = 9.

feeding ants that were exploiting the 1M food source was higher for one-entrance nests than for two-entrance nests. Respectively, around 80% and 60% of the total ant population were present on the richest food source (Fig 6C).

Furthermore, in the case of two-entrance nests, the level of selection of the best food source, i.e. the proportion of feeding ants located at the 1M food source at the end of the experiment, was significantly correlated to the level of asymmetry in the outflows of ants at each entrance (Spearman's correlation, $r = 0.83$, $n = 9$, $p = 0.005$, Fig 7). This indicated a stronger selection of the most rewarding resource when the outgoing foragers exited preferentially from one of the two entrances during the first steps of recruitment.

Once foragers had reached the feeders, they showed a higher propensity to drink at a more concentrated sugar solution. For the one-entrance nest condition, ants that reached a feeder were twice as likely to drink at the 1M than at the 0.1M food source (0.74 and 0.36 for the 1M and 0.1M source respectively, $n_{1M} = n_{0.1M} = 35$). These ants also stayed on average three times longer at the 1M than at the 0.1M source (144±98 seconds and 52±42 seconds for the 1M and

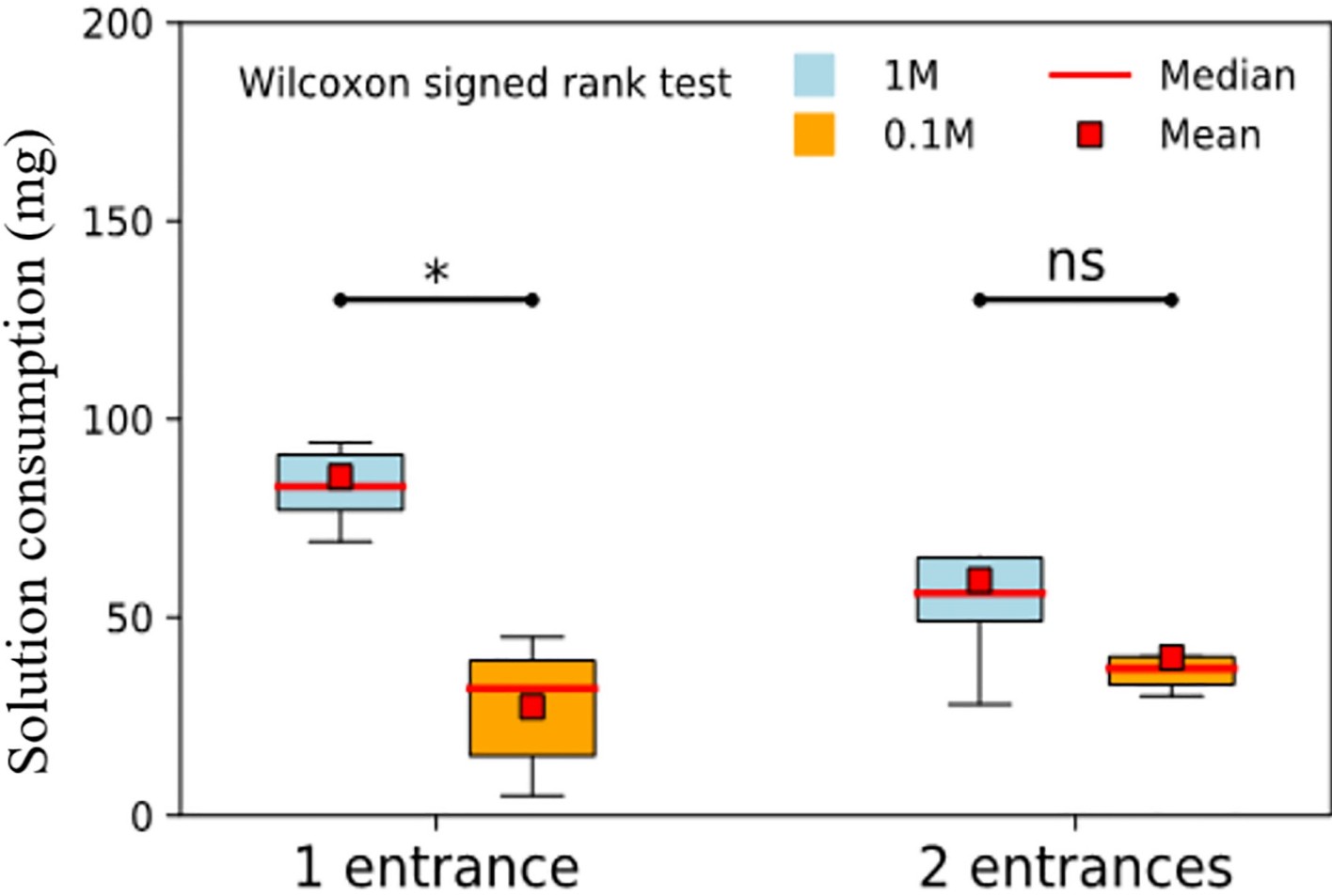

**Fig 8. Total amount of sucrose solution retrieved from the 1M and 0.1 feeders in each nest entrance configuration.** We measured the ingested amount of sucrose solution at the end of the experiment for one- and two-entrance nests. Blue and orange boxplots represent the food solution consumption at 1M and 0.1 M feeders respectively. Presented are medians and quartiles, red squares indicate means and circles indicate outliers (n = 9, Wilcoxon signed rank test).

0.1M source respectively, $n_{1M} = n_{0.1M} = 35$). The addition of a second entrance did not alter these feeding behaviours. Indeed, ants showed the same probability to start drinking regardless of nest configuration when being at the 1M feeder ($p_{1ent} = 0.74$ and $p_{2ent} = 0.55$ for the one-entrance and two-entrance nest respectively, $n_{1ent} = n_{2ent} = 35$, Chi square test, $p = 0.08$) or when being at 0.1M feeder ($p_{1ent} = 0.36$ and $p_{2ent} = 0.43$ for the one-entrance and two-entrance nest respectively, $n_{1ent} = n_{2ent} = 35$, Chi square test, $p = 0.62$). Ants also spent a similar feeding duration of 144±98 and 131±140 seconds at the 1M food source (Mann-Whitney U test, $n_{1ent} = n_{2ent} = 35$; $p = 0.12$), and of 52±42 and 74±96 seconds at the 0.1M source (Mann-Whitney U test, $n_{1ent} = n_{2ent} = 35$, $p = 0.86$) in one- and two-entrance nests respectively. These results suggest that the individuals that were mobilized out of a one-entrance or a two-entrance nest did not differ in their feeding motivation, once they had reached the food source. At the colony level, ants ingested a similar total amount of sugar solution, regardless of the nest configuration, with 112±20 mg and 99±26mg of food solution being retrieved in one-entrance and two-entrance nests respectively (mean±SD, Wilcoxon signed rank test, $n = 9$, $p = 0.12$ Table 1; Fig 8). However, the 1M sucrose solution represented more of the total amount of ingested food, for colonies kept in one entrance nests (76% on average, $n = 9$) than for colonies

kept in two-entrance nests (60% on average, $n = 9$). This resulted in a significantly higher amount of the most concentrated food solution being retrieved in nests with a single entrance than in two-entrance nests (Wilcoxon signed rank test, $n = 9$, $p = 0.019$, Table 1, Fig 8). When converting the values of ingested sugar solution into the corresponding amount of sucrose carbohydrates that was retrieved by foragers, colonies housed in one-entrance nests benefited from higher energetic incomes than two-entrance-nest colonies (mean±SD, 30.0±5.2 mg and 21.5±7.1 mg of sucrose respectively, Wilcoxon signed-rank test, $p = 0.027$, Table 1). In terms of foraging efficiency, when taking into account the higher mobilisation of workers in two-entrance nests, the sugar yield per mobilised ant was more than twice higher in one-entrance than in two-entrance nests ($n = 7$, Wilcoxon signed rank test, $p = 0.016$, Table 1). Overall these results suggest that, although the mobilization of foragers increased in two-entrance nests, multiple entrances led to a decreased ability of ants to collectively select and exploit the most rewarding resource.

## 5. Discussion

This study demonstrates that the structure of the nest-environment interface influences collective decision-making by ants. Adding a second entrance to the nest appeared to reduce the efficiency of information sharing between foragers and to hamper their ability to collectively select the best available resource. Although an additional entrance allowed for the recruitment of twice as many nestmates, a smaller proportion of workers actually reached the food sources and were distributed more evenly between food sources regardless of their sugar concentration. Multiple entrances thus resulted in a lower foraging efficiency and a lower amount of carbohydrates that were ultimately retrieved inside the nest.

In many social species such as ants, the coupling of interactions between nestmates with positive feedback loops, favours the emergence of collective strategies of food exploitation. In mass recruiting ants such as *M. rubra*, these amplifying processes are based both on direct contacts, such as antennations and trophallaxis taking place at the nest entrance [29–31, 32, 34, 36], and on indirect interactions, via pheromone trails laid outside the nest [31,37,38]. In the present study, where two food sources were available in the environment, the level of ants' mobilisation out of two-entrance nests doubled compared to one-entrance nests. Most probably, two-entrance nests allowed recruiters to come into contact with a larger audience of potential foragers than one-entrance nests, which could have favoured the exit of twice as many recruits. Similarly, in the pioneering Pinter-Wollman study, a highly connected entrance chamber, which increases the number of locations where ants can be recruited, enhances the dynamics of mobilization of foragers to food [39]. Interestingly, in a previous study [31, S1 Table], where a single food source was present in the environment, the global mobilisation of workers was found to be similar in both one- and two- entrances nests. A plausible explanation is that, with only one food source, recruitment was downregulated, due more encounters among foragers on the path [40] and at the food source [41]. When compared to the one feeder/one entrance condition [31], the highest ant mobilization observed in the case of two-feeders/two entrances could thus result both from a wider audience of potential recruits located near the two entrances and from the spacing of several food sources over a wider area, which increases the likelihood for ants to discover food and reduces the downregulating effects of crowding on recruitment.

Two-entrance nests enhanced the global mobilisation of workers but, at the same time, there was a decrease in the efficiency of individual foragers to reach the food target, even once the recruitment was well established. Likewise, in the case of a single food source [31, S1 Table], multiple nest entrances make the foraging trail less likely to emerge between the nest

and the food source and the recruits less likely to reach the food source. This indicates that the second component of the recruitment process, i.e. the guiding role of the pheromone trail, is less efficient when the nest had multiple entrances. Indeed, the global direction that the ants follow while they are heading toward the food source or while they come back to the nest, is provided by the trail pheromone laid by successful foragers (see e.g. [37,38]) as well as by home-range marks laid near the nest entrance [42–44]. In the case of several food sources and/ or nest entrances, ants are faced with multiple possible paths that are connecting the nest to available resources. This may increase their probability to lose track of a foraging trail and/or may prevent them from orienting along a well-defined gradient of area marking, thereby leading to a lower efficiency of foraging journeys.

At the collective level, an additional entrance, through which information could transit, decreased the efficiency of social foraging and ultimately led to a lower amount of retrieved food [31, S1 Table]. Furthermore, when an ant colony was faced with two food sources of different quality, the current study demonstrates that multiple entrances hampered the selection of the most rewarding resource. The proportion of ants exploiting the best resource continuously increased in one-entrance-nests, until reaching 80% of the foragers' population, a value also found in other mass recruiting ant species like *Lasius niger* [45]. By contrast, when housed in two-entrance nests, foragers distributed themselves more evenly, with the rich food source attracting only around 60% of the foragers. Occasionally, a selection of the richest resource could be observed when ants favoured the use of only one of the two entrances and thus exchanged information at a single location. The poor selection of the best resource, coupled to the larger number of ants mobilized out of two-entrance nests, resulted in an energetic yield per forager that was 2.5 times lower in two-entrance nests than in one-entrance nests. Any random event (e.g. a delay in the time of food discovery) coupled to amplifying phenomena (e.g. the laying of a recruitment trail) may lead to the selection of a resource of a poor quality over a richer food source [9,45,25]. Theoretical studies also suggest that the number of options increases possible irrationalities in decision-making and influences the overall quality of the decision [46]. In the present study, we demonstrate that accurate collective choices and foraging efficiency also depend on the convergence of successful scouts at a single entrance, which allows naive workers to compare trails of different intensity leading to food sources of different quality. Through this process of competing positive feed-backs, the most concentrated trail will be the most likely to attract nestmates, its recruiting signal will be further reinforced by the mobilized foragers and ultimately the whole colony will collectively focus its foraging activity on the most rewarding source [25]. Likewise, in the case of group-leading coupled to mass recruitment, as observed in *Tetramorium caespitum* ants [47], a centralization of competing recruiters allows potential recruits to encounter mutually exclusive leaders, what will facilitate the collective selection of the most rewarding resources. By segregating recruitment stimuli at several distinct locations inside the nest, multiple entrances disrupt the ability of nestmates to compare alternative information and jeopardize the collective selection of the most rewarding food target. This results in less accurate foraging decisions and in a potential loss of energetic incomes for the whole colony. At the extreme, for large angles between food sources or for nests with more distant entrances "behaving" as separate cavities, a comparison of incoming information and a collective selection of the most rewarding source might no longer take place since nest entrances would be activated by their own recruitment process and recruitment trails exiting from nest holes would be more spatially distinct. Finally, building a consensus on different options and selecting the most valuable one can be time consuming, particularly in a system of shared decision-making as observed in many insect colonies. Distributing incoming information between several locations may prevent the reaching of a consensus within a realistic time frame. Such a delay of decision-making appears particularly detrimental when social

insects need to use a collaborative strategy to exploit food resources and to monopolize them against competitors [25].

If multiple entrances counteract the ants' ability to discriminate between resources of different quality, they can nonetheless provide some advantages to the colony by diversifying the foraging zones travelled and explored by outgoing ants. As for polydomic ant nests, albeit to a smaller spatial scale, multiple entrances can decrease the distance foragers have to travel in the outside before reaching resources, reduce the energetic costs of food collection and provide shelters to foragers limiting their risks of being predated [48–53]. Furthermore, while being at the expenses of an efficient food selection, a multiplicity of nest entrances results in a more homogeneous distribution of foragers between available resources [54], what would enhance the robustness of the whole colony to fluctuations–and possible depletion–of exploited resources. This may be especially profitable in the case of a moderately opportunistic ant such as *M. rubra*, that feeds on both stable resources such as aphids' honeydew, but also small scattered insect corpses (personal observations).

Put in a wider ecological perspective, as for the topology of nest chambers [39], studying the structure of nest interface with the outside environment, in particular its number of entrances, provide insights into the processes that regulate information sharing and collective strategies of resource exploitation. Now, the question is whether there is a correspondence between the "permeability" of the nest interface, i.e. the number of nest entrances, and the relevant properties of the outside environment including its stability, the distribution of resources and the costs of threats. Further studies should investigate to which extent the nest-environment interface is an adaptive structure that fits to the decision-making processes of the inhabiting ants as well as to the specificities of the resources at stake.

## Supporting information

**S1 Table. Impact of multiple nest entrances on ants' foraging towards either a single feeder (1M sucrose solution) or two feeders of different quality (1M Vs 0.1M sucrose solution).** The table lists the main findings of the current paper (Two feeders) and of a previous paper by Lehue et al 2020 (One feeder) that used an identical experimental setup but different ant colonies. A positive or a negative sign means that the foraging characteristics is respectively favored or hampered by the opening of a second nest entrance. A sign put between brackets means that only a trend (not statistically significant) was observed. 0 means that no impact was found. NA: Not available data due to the lack of well-defined trail over the foraging area (Two feeders) or the lack of opportunity of food choice (One feeder).
(DOCX)

## Acknowledgments

We thank Luc Dekelver for helping to collect ant colonies and Dr Collignon for creating the final version of figures. Comments from Dr Czaczkes and an anonymous referee helped to substantially improve the manuscript.

## Author Contributions

**Conceptualization:** Marine Lehue, Claire Detrain.

**Data curation:** Claire Detrain.

**Formal analysis:** Marine Lehue, Claire Detrain.

**Investigation:** Marine Lehue.

**Methodology:** Claire Detrain.

**Project administration:** Claire Detrain.

**Writing – original draft:** Marine Lehue, Claire Detrain.

**Writing – review & editing:** Claire Detrain.

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
