## [Decision Letter · Decision Letter 0]

8 Apr 2020

PONE-D-20-06920

More nest entrances affect collective decision-making and foraging efficiency in
ants.

PLOS ONE

Dear Dr. Detrain,

Thank you for submitting your manuscript to PLOS ONE. After careful consideration, we
feel that it has merit but does not fully meet PLOS ONE’s publication criteria as it
currently stands. Therefore, we invite you to submit a revised version of the
manuscript that addresses the points raised during the review process.

Both
reviewers raise very useful points about the relevant published literature, data
analysis and the arguments/conclusions. I would appreciate if you could answer as
many of these concerns as possible using the data you have already collected. If a
question requires additional data collection (and I assume this is not currently
possible because of the shutdown), please state so in the response to
reviewers.

We would appreciate receiving your revised manuscript by May 23 2020 11:59PM. When
you are ready to submit your revision, log on to https://www.editorialmanager.com/pone/ and select the 'Submissions
Needing Revision' folder to locate your manuscript file.

If you would like to make changes to your financial disclosure, please include your
updated statement in your cover letter.

To enhance the reproducibility of your results, we recommend that if applicable you
deposit your laboratory protocols in protocols.io, where a protocol can be assigned
its own identifier (DOI) such that it can be cited independently in the future. For
instructions see: http://journals.plos.org/plosone/s/submission-guidelines#loc-laboratory-protocols

We look forward to receiving your revised manuscript.

Kind regards,

Olena Riabinina, PhD

Academic Editor

PLOS ONE

"M.L. was supported by a Belgian PhD Grant from the F.R.I.A. (Fonds pour la formation
à la Recherche dans l’Industrie et dans l’Agriculture). C.D. is Research Director
from the Belgian National Fund for Scientific Research (FRS-F.N.R.S)."

We note that one or more of the authors are employed by a commercial company: Belgian
National Fund for Scientific Research

4. Your ethics statement must appear in the Methods section of your manuscript. If
your ethics statement is written in any section besides the Methods, please move it
to the Methods section and delete it from any other section. Please also ensure that
your ethics statement is included in your manuscript, as the ethics section of your
online submission will not be published alongside your manuscript.

Reviewers' comments:

Reviewer's Responses to Questions

Comments to the Author

1. Is the manuscript technically sound, and do the data support the conclusions?

Reviewer #1: Yes

Reviewer #2: Yes

2. Has the statistical analysis been performed
appropriately and rigorously? 

Reviewer #1: Yes

Reviewer #2: Yes

3. Have the authors made all data underlying the
findings in their manuscript fully available?

Reviewer #1: Yes

Reviewer #2: Yes

4. Is the manuscript presented in an intelligible
fashion and written in standard English?

Reviewer #1: Yes

Reviewer #2: Yes

5. Review Comments to the Author

Reviewer #1: In this manuscript, Lehue and Detrain continue their exploration of the
role of the nest entrance on ant foraging. Specifically, they ask how having one or
two nest entrances affects the recruitment and foraging dynamics of Myrmica rubra
colonies foraging on two sucrose feeders of different qualities (0.1 vs 1M. They
report that while having two nest entrances causes a larger surge of recruitment to
newly discovered food sources, foraging efficiency suffers for multiple reasons. Key
amongst them is that foraging is less well-focussed on the high quality feeder. In
addition, fewer recruits successfully find the food source within 10 minutes of
leaving the nest.

I found this work interesting, well carried out, and generally well written. The
methodologies used were appropriate, and I could detect no critical methodological
issues. Almost all of the conclusions drawn are well supported by the data (for one
exception, see below). In all, this is a good contribution in an understudied area,
and I recommend publication with moderate revisions. However, some key literature is
not discussed, and there are methodological choices which may limit the
generalisability of the results. Moreover, one of the main results (increased
recruitment in 2-entrance nests) did not replicate in the authors previous paper,
and the explanation for this was not fully convincing. I also have a series of more
minor points about the figures, the supplement, and some grammatical suggestions. I
hope the extensive comments I have made show how much I value this work.

Introduction

The introduction is rather long and generic, with the whole first page and a half
being basically suitable to any collective decision-making paper on ants. One
example is lines 51-56. The authors might consider cutting this generic aspect of
the introduction down. They will need this for the next point:

Two papers are key to understanding this work. The first is a previous recent paper
by the authors (Lehue et al. 2020), and the second is Pinter-Wollman (2015). The
previous work needs to be discussed in much more detail – specifically, all of the
main results of that experiment need to be listed, and it should be mentioned that
the experiments were identical except for the food sources used. Indeed, the
discussion might even benefit from a small table listing the main findings of the
current paper and this previous paper, so that we can ‘see’ this body of work in one
place.

I was surprised that the Pinter-Wollman paper was not mentioned at all, considering
its deep parallels to the current work. Specifically, the paper shows that as
connectivity increases, so does recruitment speed. This is very similar to what is
reported in the current paper. This paper should thus be discussed in some detail
both in the introduction and in the discussion.

Ln 33 – information sources

Ln 33-34 – some of these references are not about social insect decision making, but
the sentence is only about that.

Ln 91 – kept, not hosted.

Methods

The food sources were rather close to each other. The resultant pheromone trails from
the multiple nest entrances thus cross over quite extensively. I wonder, if the
angle between the food sources and the nest were larger (at the extreme, if the food
sources were at each side of the nest), whether the effects described in this paper
would be present at all. This is probably an issue worth raising in the discussion –
how generalisable are these results?

A second methodological concern is the size of the nest: these are quite small nests
(300 workers, 8cm diameter), and the distance between the two entrances is quite
small (c. 3cm, right?). Thus, as the author notes, the ‘activation area’ of the two
nest entrances overlap, possibly leading to increased recruitment. However, in a
natural nest, which covers more ground and is composed of a tunnel system, multiple
nest entrances likely do not overlap in effect. Again, it is important to ask: how
realistic and generalisable are the results, given these methods? I would argue
that, if the nest were composed of a tunnel system, and the two nest entrances were
in two somewhat separate cavities (as likely occurs in nature), very different
dynamics might occur. This does not mean that the current experiment is meaningless
– far from it – but it does suggest that we don’t have the whole story here (see the
Pinter-Wollman paper, for example). I think this issue needs to be raised
explicitly. Maybe in the future, this experiment could be repeated with a nest
composed of tunnels and cavities…

Ln 198 – define “ants present at a food source” more formally. Touching the food
source? Within 1cm of the food source?

Results

Table 1 – while later in the manuscript it is claimed that the amount of sucrose
taken in by colonies is given (ln 370-373), this does not seem to be what is
reported here. I note, for example, that 26.1 + 0.8 = 26.9 (column 3). This suggests
that the 0.1M sucrose has not, in fact, been reduced by a factor of 10 to account
for having only 10% as much sucrose as 1M. The key foraging metric here, after all,
is not weight of sucrose solution returned, but the weight of sucrose returned. 1
molar sucrose has ten times as much sugar as 0.1M. Please amend this, and rerun the
analysis, if necessary. It will only strengthen the results.

A large concern for me was that one of the main results – the increase in forager
mobilisation from 1 to 2 nest entrances – was not replicated in Lehue et al. (2020)
(discussed in ln 409-411). This does not make much sense to me, and the hypothesis
proposed by the authors is unconvincing. They argue, if understood them correctly,
that more food sources would mean a higher rate of returning workers, so higher
recruitment and activation. Firstly, given the cross-shaped feeders and presumably
no queuing at the feeders, this doesn’t sound likely. Secondly, it is not clear to
me why have one or two nest entrances should affect the total number of returning
foragers, regardless of the number of food sources. One possibility is that, with
only one food source or only one entrance, recruitment is downregulated, due more
encounters with ants on the path (Czaczkes et al. 2013) and at the food source
(Wendt et al. 2020). However, luckily the authors have all the data needed to
support or reject their hypothesis in their videos. If they were to count the number
of returning foragers (assessable by abdomen distention) within the first 10
minutes, their hypothesis predicts many more returning foragers in the two-entrance
configuration. Don’t forget to collect this data blind to treatment, though, as
abdomen distention is quite hard to assess in M. rubra! As an aside, examining the
raw data from this and the previous paper, I note that that the foraging dynamics of
the colonies are very similar for one food source one entrance (previous paper), one
food source two entrances (previous paper), and two food sources two entrances (this
paper), with the two food sources two entrances standing out, with double the
recruitment. This suggests there is something special about this configuration – it
might be worth presenting the data from the 2020 paper again here for
comparison.

Ln 468 – replace “dispatching of multiple information” with “distributing incoming
information”

Ln 498 – replace “to draw” with “for creating”

Supplement - data

I applaud the authors for providing the raw data. All papers should do this. However,
the data is presented in a very hard to use format in the supplement. Firstly, all
labels are in French – this should be amended. Secondly, the data has not been
entered in a ‘tidy’ manner – tidy having a specific definition in data entry
(https://vita.had.co.nz/papers/tidy-data.html). In short – every
column should be a variable, and every row an observation. Please also provide
metadata (information about what each column is). This all may sound terrible and
pedantic (I apologise!), but it really is very important, and will facilitate data
analysis and visualisation for you in the future.

Figure 2, 3, 4A&B– please add mean connect lines. Indeed, this data is not
normally distributed, so you should probably be using medians and quartiles, not
means and SD.

Figure 4 A and B – These figures make seeing the main comparison (1 vs 2 nest
entrances) very difficult, and are more appropriate for exploring the dynamics,
which is a side issue here. Perhaps just present the total mean time for one and two
nest entrance? Indeed, just mean proportion ants at feeder (two bars) captures
everything we need to know, but a figure similar to figure 8 would also be ok.

Figure 4C – why does this now have connection lines but no error ribbons?

REFERENCES CITED

Czaczkes TJ, Grüter C, Ratnieks FLW (2013) Negative feedback in ants: crowding
results in less trail pheromone deposition. J R Soc Interface 10:. https://doi.org/10.1098/rsif.2012.1009

Lehue M, Collignon B, Detrain C Multiple nest entrances alter foraging and
information transfer in ants. Royal Society Open Science 7:191330. https://doi.org/10.1098/rsos.191330

Pinter-Wollman N (2015) Nest architecture shapes the collective behaviour of
harvester ants. Biology Letters 11:20150695. https://doi.org/10.1098/rsbl.2015.0695

Wendt S, Kleinhölting N, Czaczkes TJ (2020) Negative feedback: Ants choose unoccupied
over occupied food sources and lay more pheromone to them. Journal of The Royal
Society Interface. https://doi.org/10.1098/rsif.2019.0661

Reviewer #2: I found this to be a carefully conducted study. It shows that ants are
less capable of choosing a richer food source over a weaker one, when the ants have
more than one exit hole through which they can leave their nest. Two related things
concerned me about the study, which may well be answerable.

1. The short time period during which it seems that the colonies were examined. Do
ants improve if they are allowed to forage freely for a few days with two entrances
before being tested? You say they were allowed ad lib food for a time with two
entrances but without specifying the details. Would be nice to find out if ants do
improve over time, by, for instance, coming to use just one of the exits.

2. Could the lack of adaptation to the problems that you set in part be a consequence
of abnormal conditions in the artificial nests during testing? E.g. they haven't
time to sort themselves out?

Line by line commentary

Line 78. Somewhere around here you need a bit of natural history about nest
entrances. There are obvious examples of nests with one entrance (e.g. Cataglyphis
fortis). Are there any with several exit holes (Formica rufa ?) and in such cases do
individual ants always use the same hole to enter or exit by same hole or do they
have other methods of successful recruitment?

Line 100. Be nice to know rough numbers of ants in wild colonies to compare with the
300 in your experimental ones.

Line 118. Do you have evidence that 48 hours is long enough for ants to be
acclimated? Might they organise themselves to cope with 2 entrances if given a
longer period?

Line 140. Did ants have the chance to explore the arena during this ad lib period or
was the food adjacent to the nest and exploration prevented by a barrier? Relevant
to general comment at the start of these comments.

Line 240. Was there any correlation between asymmetry of entrance use and proportion
of visits to the high concentration feeder?

Line 272. Do you have proportions of how many ants in the one and two nest conditions
returned after feeding and how many returned empty stomached?

Line 282 Fig. 4 would be helpful if you label in legend and fig the three conditions
(legend) and arrows (fig) as 'a', 'b' or 'c'.

Line 291. I'm curious whether you videoed the foraging and if so whether you can say
anything about trail use in the two entrance conditions.

Line 293 'as the 0.1' not 'than the 0.1'

Line 299 Again question about asymmetry of entrance use and proportion of ants
reaching 1M sucrose. P = 0.053 doesn't mean no preference but just a weak
preference.

Line 305. Don't like 'repartiton' prefer something like 'how ants distributed
themselves between..'

Line 333. So there is a correlation between asymmetry of entrance use and food
preference. Be helpful to a reader to mention it briefly earlier - around line
240.

Line 348. suggest: 'drank about twice as much from the 1M than from the .1M
feeder'

Line 394.409 'twice as many nestmates'

Line 419 -425. I'm uncertain about the argument made here. You need more detail to be
convincing that in this experiment the outflow of foragers depends highly on
encounters between returning ants and potential foragers. Are there sufficient
returning foragers to generate the timing of maximum outflow in Fig 2 when there are
two exits? And do returning ants return to their exit hole or are they
indiscriminate? Have you compared outflow from one and two hole nests when holes are
opened after a period of being shut and there is no food to find. Does the total
peak then differ between the two nests? Para needs a conclusion of what the answer
might be.

Line 430 'at the same time'

6. PLOS authors have the option to publish the peer
review history of their article (what does this mean?). If published, this will
include your full peer review and any attached files.

If you choose “no”, your identity will remain anonymous but your review may still be
made public.

**Do you want your identity to be public for this peer review?** For
information about this choice, including consent withdrawal, please see our
Privacy Policy.

Reviewer #1: Yes: Tomer J. Czaczkes

Reviewer #2: No

---

## [Author Response · Author response to Decision Letter 0]

12 May 2020

Please see the attached document for all detailed responses to reviewer comments.
Thanks

---

## [Decision Letter · Decision Letter 1]

28 May 2020

Foraging through multiple nest holes: an impediment to collective decision-making in
ants

PONE-D-20-06920R1

Dear Dr. Detrain,

We are pleased to inform you that your manuscript has been judged scientifically
suitable for publication and will be formally accepted for publication once it
complies with all outstanding technical requirements.

With kind regards,

Olena Riabinina, PhD

Academic Editor

PLOS ONE

Additional Editor Comments (optional):

Reviewers' comments:

Reviewer's Responses to Questions

**Comments to the Author**

1. If the authors have adequately addressed your comments raised in a previous round
of review and you feel that this manuscript is now acceptable for publication, you
may indicate that here to bypass the “Comments to the Author” section, enter your
conflict of interest statement in the “Confidential to Editor” section, and submit
your "Accept" recommendation.

Reviewer #1: All comments have been addressed

2. Is the manuscript technically sound, and do the data
support the conclusions?

Reviewer #1: Yes

3. Has the statistical analysis been performed
appropriately and rigorously? 

Reviewer #1: Yes

4. Have the authors made all data underlying the
findings in their manuscript fully available?

Reviewer #1: Yes

5. Is the manuscript presented in an intelligible
fashion and written in standard English?

Reviewer #1: Yes

6. Review Comments to the Author

Reviewer #1: The authors did a good job addressing all the comments. This is a good
piece of work, and I look forward to seeing it published.

7. PLOS authors have the option to publish the peer
review history of their article (what does this mean?). If published, this will
include your full peer review and any attached files.

If you choose “no”, your identity will remain anonymous but your review may still be
made public.

**Do you want your identity to be public for this peer review?** For
information about this choice, including consent withdrawal, please see our
Privacy Policy.

Reviewer #1: Yes: Tomer J. Czaczkes

---

## [Editor Report · Acceptance letter]

8 Jun 2020

PONE-D-20-06920R1 

Foraging through multiple nest holes: an impediment to collective decision-making in
ants 

Dear Dr. Detrain:

I'm pleased to inform you that your manuscript has been deemed suitable for
publication in PLOS ONE. Congratulations! Your manuscript is now with our production
department. 

Kind regards, 

on behalf of

Dr. Olena Riabinina 

Academic Editor

PLOS ONE